# Key Residues Affecting Binding Affinity of *Sirex noctilio* Fabricius Odorant-Binding Protein (SnocOBP9) to Aggregation Pheromone

**DOI:** 10.3390/ijms23158456

**Published:** 2022-07-30

**Authors:** Enhua Hao, Yini Li, Bing Guo, Xi Yang, Pengfei Lu, Haili Qiao

**Affiliations:** 1Key Laboratory for Silviculture and Conservation of Ministry of Education, Beijing 100083, China; hao452115308@126.com (E.H.); lynn1n2021@163.com (Y.L.); guobing618@126.com (B.G.); yangxi1997000@163.com (X.Y.); 2Institute of Medicinal Plant Development, Chinese Academy of Medical Sciences and Peking Union Medical College, Beijing 100193, China

**Keywords:** *Sirex noctilio*, odorant-binding proteins, aggregation pheromone, fluorescence binding assay, computational simulation, site-directed mutagenesis

## Abstract

*Sirex noctilio* Fabricius (Hymenoptera Siricidae) is a major quarantine pest responsible for substantial economic losses in the pine industry. To achieve better pest control, (*Z*)-3-decen-ol was identified as the male pheromone and used as a field chemical trapping agent. However, the interactions between odorant-binding proteins (OBPs) and pheromones are poorly described. In this study, SnocOBP9 had a higher binding affinity with Z3D (Ki = 1.53 ± 0.09 μM) than other chemical ligands. Molecular dynamics simulation and binding mode analysis revealed that several nonpolar residues were the main drivers for hydrophobic interactions between SnocOBP9 and Z3D. Additionally, computational alanine scanning results indicated that five amino acids (MET54, PHE57, PHE71, PHE74, LEU116) in SnocOBP9 could potentially alter the binding affinity to Z3D. Finally, we used single-site-directed mutagenesis to substitute these five residues with alanine. These results imply that the five residues play crucial roles in the SnocOBP9-Z3D complex. Our research confirmed the function of SnocOBP9, uncovered the key residues involved in SnocOBP9-Z3D interactions, and provides an inspiration to improve the effects of pheromone agent traps.

## 1. Introduction

Insects use their olfactory system to sense odor changes in the environment and subsequently adjust behaviors such as mating, host positioning, food selection, interspecies recognition, intraspecies communication and habitat selection [1,2,3,4,5,6]. Chemical signals are converted into electrical signals and transmitted to the antennal lobes to activate the central nervous system of the brain, producing behavioral instructions for insects to respond accordingly [7].

Odorant-binding proteins (OBPs) are key components of the insect olfactory system and are responsible for docking with chemical molecules in the olfactory signaling transduction pathway, which was first discovered in the trichoid sensillum from the antennae of *Antheraea polyphemus* [8]. Later, OBPs were identified as a type of small water-soluble protein [9]. These proteins are generally composed of 135–220 amino acids. An obvious characteristic of OBPs is that the amino acid sequence contains six cysteine sites and forms three pairs of disulfide bonds to ensure structural stability. Studies have also shown that some OBPs have “plus-c” or “minus-c” types, which contain more or less than six cysteines, respectively [10]. According to the types of odor volatiles they bind, OBPs can be divided into general odor binding proteins (GOBPs) and pheromone binding proteins (PBPs) [11].

OBPs are currently considered to have multiple functions, especially when OBPs are expressed in non-olfactory tissues. For instance, a new study in *Bombyx mori* showed that the investigated OBPs are expressed in non-olfactory tissues at different developmental stages, strongly indicating that these OBPs have multiple functions in addition to olfactory functions [12]. Through Western blot analysis and immunofluorescence localization, AaegOBP22 was found to be expressed in male sex organs in *Aedes aegypti*, and to have characteristics of selective binding with various ligands [13]. Similar findings suggest that OBPs may have multiple functions [14,15,16,17]. However, for the odorant-binding proteins that were abundantly expressed in the antennae, most relevant studies still indicate that OBPs were more inclined to reflect their odorant-binding function. To date, a large number of OBPs have been discovered and their binding affinity to odorant ligands has been investigated via fluorescence competitive binding experiments. Most of these studies focused on Lepidoptera. For example, AipsGOBP1 (*Agrotis ipsilon*) [18], CpunGOBP1 (*Conogethes punctiferalis*) [19], SlitPBP2, SlitPBP3 (*Spodoptera litura*) [20] and EphipPBP1 (*Eogystia hippophaecolus*) [21] can bind to either their host plant volatiles or pheromones. The same techniques were used to study the binding affinity of the OBPs of insects in other orders to different kinds of chemical ligands. BodoOBP1 and BodoOBP2 (*Bradysia odoriphaga*) in Diptera [22]; CforOBP1, CforOBP2 and CforOBP3 (*Cylas formicarius*) in Coleoptera [23] as well as TjapOBP2 (*Trissolcus japonicus*) [24], McinNPC2 (*Macrocentrus cingulum*) [25] and AcerOBP15 (*Apis cerana cerana*) [26] in Hymenoptera can also bind with either general plant volatiles or pheromones. However, the mechanisms underlying the interactions between OBPs and ligands are still unclear.

With the development of computer technology, the molecular dynamics (MD) method of protein–ligand analysis has been applied in many fields, such as medicine, materials science and computational chemistry [27,28,29]. Altering the affinity of OBPs to small ligands by replacing specific residues in the binding pocket may lead to several technological applications. Thus, a growing number of studies have uncovered the binding modes of some insect proteins and important ligands through computational simulations and site-directed mutations of key amino acids. For instance, the results of MD simulation for MvicOBP3 (*Megoura viciae*) and the aphid alarm pheromone, (*E*)-β-farnesene, showed that MvicOBP3 has a highly hydrophobic interaction with the ligand [30]. Fluorescence quenching and MD simulation experiments revealed that TRP106, a key amino acid in AmalOBP8 from *Agrilus mali*, played a very important role in the formation of hydrogen bonds with geranyl formate [31]. Therefore, according to reverse chemical ecology, OBPs can be used as a major entry point for the development of new and environmentally friendly attractants once the key amino acid residues and odor ligands for binding are determined [32].

*Sirex noctilio* Fabricius (Hymenoptera: Siricidae) is a major international quarantine pest native to southern Europe, North Africa, and parts of Asia [33]. Since its discovery in New Zealand in 1900, it has been successively found in Australia, Uruguay, Argentina, Brazil, South Africa, Chile, the United States, Canada and further South American countries [34,35,36,37,38]. *S. noctilio* was found in Daqing City, Heilongjiang Province, China, in 2013 and was listed as a national forestry hazardous pest in China the following year [39,40]. It mainly harms coniferous species and leads to significant economic losses in invaded cities, but also causes immeasurable damage to the local ecological environment [41]. Thus, there is an urgent need for developing a new type of chemical trapping agent to monitor and suppress the pest population.

Attractant-baited intercept traps are widely used for the management of forest insect pests, and both pheromones and plant volatiles are commonly used to develop attractants. Pheromones are commonly used as attractants to evaluate pest populations, and several pheromone compounds for *S. noctilio* are known [42,43]. Female pheromones derived from cuticular washes were identified as (*Z*)-7-heptacosene, (*Z*)-7-nonacosene and (*Z*)-9-nonacosene. These pheromones were found to be active in laboratory assay experiments and were posited to be short-distance contact pheromones [44]. A male-produced pheromone was identified as (*Z*)-3-decen-ol (Z3D) and found to produce a strong response in both male and female antennae with gas chromatography–electroantennographic detection (GC-EAD) assays. In addition, Z3D was found to be the most active substance in Y-tube olfactometer and wind tunnel assays. Several putative or suspected minor components, including (*Z*)-4-decen-ol and (*E, E*)-2,4-decadienal, were not identified with GC-MS and were only active in behavioral tests [42]. These results are consistent with our previous experiments in a Chinese population. Only Z3D was identified from the *S. noctilio* Chinese population, and the peak of Z3D release occurred from 11:00–12:00 for two-day-old individuals, which can elicit antennal response consistently in our GC-EAD assays (Appendix A) [45,46,47]. Two putative male aggregation pheromones in *S. noctilio* that have not been validated in field-activity were identified as (*Z*)-3-octen-ol and (*Z*)-3-dodecen-ol; these two new substances were not identified in our previous experiments [48]. This pheromone-related research indicated that Z3D plays an important role in regulating the behavioral rhythm of woodwasps. Secondly, kairomone (plant volatiles) lure traps were found to be the most effective method for trapping *S. noctilio* individuals in areas with large populations, and (+)-Limonene, (−)-Limonene, Alpha-Pinene, Beta-Pinene, 3-Carene, Camphene were attractive host plant volatiles in early reports [49,50,51,52]. At present, plant volatile compounds baited with black panel or multiple-funnel traps are the dominant traps used for detecting and monitoring *S. noctilio*. However, these traps have relatively good effects with moderate to large populations, but are ineffective with small populations, especially in environments with competitive volatiles produced by the host [53]. Finally, females carry arthrospores of a specific fungus, *Amylostereum areolatum*, in specialized internal organs called mycangia. Both fungus and mucus are synchronously injected into oviposition holes through the bark along with the eggs [54]. It is speculated that the volatiles of symbiotic fungi may be linked to flying behaviors such as courtship or habitat finding. Previous research demonstrated that the volatiles of *A. areolatum* ((–)-Globulol, 2-hexene, cycloprop[e]indene-1a,2(1H)-dicarboxaldehyde, terpene and cyclopentanone) were highly attractive to female woodwasps [55].

In previous studies, sixteen *SnocOBPs* genes were obtained based on the antennae transcriptome data and quantitative polymerase chain reaction (qPCR) analysis of the tissue expression of nine olfactory genes. By constructing a phylogenetic tree and comparing the OBP protein sequences of *S. noctilio* with those of Hymenoptera, Diptera and Lepidoptera, it was found that SnocOBP9 and other Hymenoptera PBP proteins were clustered in the same lineages. A qPCR experiment demonstrated that SnocOBP9 was abundantly expressed in both male and female antennae. These results indicate that SnocOBP9 may be involved in the process of chemoreception in insects [56]. However, it has not been determined whether SnocOBP9 possesses olfactory functions, and the mechanisms underlying the interactions between SnocOBP9 and ligands are also unclear [57].

In this study, the binding affinities of SnocOBP9 to three kinds of volatiles (aggregation pheromones, host plant volatiles and symbiotic fungal volatiles) were first measured using fluorescence competitive binding experiments. Afterward, based on MD simulations, per-residue free energy decomposition, computational alanine scanning (CAS), and site-directed mutation, the protein–ligand binding mode was identified, and the key amino acid residues of SnocOBP9 essential in the binding interaction with the pheromone volatile Z3D were eventually clarified. Our studies elucidate the function of SnocOBP9 and provide ideas for *S. noctilio* prevention and control from a new perspective.

## 2. Results

### 2.1. Sequence Analysis of SnocOBP9

The coding sequence was identical to the previously identified SnocOBP9 sequence in the *S. noctilio* transcriptome. The SnocOBP9 ORF was 426 bp, encoded 142 amino acids, and had a predicted size of 16.16 kDa, an isoelectric point of 4.58, and a 21 amino acid N-terminal signal peptide. We performed sequence alignment between PBPs from the other six Hymenoptera species and the SnocOBP9 sequence (Figure 1), and the results show that SnocOBP9 conformed to the PBP subfamily characteristics, with 130–150 amino acids, a signal peptide region, and six typical cysteines.

### 2.2. Expression and Purification of Recombinant SnocOBP9WT and SnocOBP9MT

After lysis and centrifugation of the cells, the recombinant proteins were mainly located in the sediment, indicating that both SnocOBP9WT and SnocOBP9MT were expressed as inclusion bodies. Thus, the recombinant proteins were purified by affinity chromatography and then denatured and renatured to obtain soluble purified proteins. To identify that the expression of the protein SnocOBP9 was successful, we performed a Western blot experiment. As the primary antibody, 6*His-Tag was used, Goat Anti-Mouse as the secondary antibody, and there is a clear target band in Figure 2A. It was proven that SnocOBP9 was successfully expressed. After removing 6*His-Tag, SnocOBP9WT (*Sirex noctilio* odorant-binding protein 9 wild-type) (Figure 2B) and SnocOBP9MT (*Sirex noctilio* odorant-binding protein 9 mutant-type) (Appendix A) produced protein bands less than 15 kDa on SDS-PAGE, consistent with the expected size.

Next, we performed Protein Digestion and LC/MS-MS identification (Appendix A) of SnocOBP9 by trypsin. Results show that three peptides (Table 1) were retrieved and assigned to the target protein (Uniprot ID: SnocOBP9, A0A857N3E7).

The three peptides existed in the protein sequences 56–68 (ADNGDIPNEQNIK, score 112.93), 42–68 (CMDEHKVDQATIDKADNGDIPNEQNIK, score 118.41), and 48–68 (VDQATIDKADNGDIPNEQNIK, score 107.33) (Table 2). This result demonstrates that SnocOBP9 had a higher abundance, proving that the expressed protein samples contained a large amount of SnocOBP9.

Finally, in order to verify the secondary structure of recombinant SnocOBP9, we conducted a structural analysis of the target protein by CD spectrum (Figure 3). The secondary structure was fitted and calculated by CDNN, and the target protein has Helix (19.6%), Beta-Antiparallel (21.50%), Beta-Parallel (l5.70%), Beta-Turn (17.70%) and Random Coil (38.70%). Through Western, LC-MS/MS and CD spectra experiments on SnocOBP9, it could be determined that the recombinant protein contained large quantities of SnocOBP9, and the renatured protein also has a corresponding secondary structure, which can be used for the next binding experiment.

### 2.3. Fluorescence Competitive Binding Assay

The isolation of the N-phenyl-1-naphthylamine (1-NPN) probe resulted in weak fluorescence on excitation at 337 nm. With the addition of the SnocOBP9WT protein, the emission spectrum of 1-NPN shifted from 480 to 420 nm, with a considerable 1-NPN-dose-dependent increase in fluorescence intensity. SnocOBP9 was titrated with increasing concentrations of 1-NPN to obtain the binding curve for 1-NPN and SnocOBP9. The Scatchard equation was calculated, and the dissociation constant value (K_1-NPN_) was Kd = 6.87 ± 0.73 μM (Figure 4A). The results indicated that 1-NPN can bind to proteins, and 1-NPN can be used as a suitable probe for subsequent experiments. Using 1-NPN as the probe, we investigated the binding affinity of SnocOBP9WT to 12 ligands (host volatiles, symbiotic fungal volatiles and insect pheromones) (Figure 4B).

The results show that SnocOBP9WT had the strongest binding affinity to aggregation pheromone volatile Z3D (Ki = 1.53 ± 0.09 μM) (Figure 5C and Table 3). SnocOBP9 also demonstrated binding to host volatiles (+)-Limonene and (−)-Limonene, but with higher Ki values (Ki = 4.87 ± 0.94 μM, Ki = 5.50 ± 0.49 μM) (Figure 5A and Table 3) and less than 50% in the range of test concentrations for symbiotic fungal volatiles (Figure 5B and Table 3). Therefore, we used the SnocOBP9-Z3D complex system for subsequent studies.

### 2.4. Protein Homology Modeling and Molecular Docking

To further investigate the interactions between SnocOBP9 and Z3D, we constructed a 3D model of the protein (Figure 6A) and chemical ligand. Sequence alignment showed that the amino acid sequence of *A. mellifera* ASP1 (PDB ID: 3cdn_A) (Figure 6A) had the highest similarity to the sequence identity of SnocOBP9 (Figure 6B), which indicates that AmelASP1 could be used as a homology modeling template for SnocOBP9.

The Ramachandran plot shows that all non-Pro/Gly residues in the protein model were trapped in allowable regions, and 94.1% of the amino acid residues were located in more preferred regions (Appendix A). Further, Verify 3D results (Appendix A) show that at least 80% of the amino acid scores were higher than 0.2. The 3D structures of AmelASP1 and SnocOBP9 show that both proteins contained seven α-helixes and one internal cavity structure, and both binding pockets are shown in Appendix A. These results indicate that the stereochemistry and compatibility of the protein model were reliable, and that the binding mechanisms of AmelASP1 and SnocOBP9 were similar. Therefore, the subsequent computational calculations were based on the SnocOBP9 homology model. Then, we obtained the best-scored (Appendix A) SnocOBP9-Z3D conformation through Autodock Vina for the next MD analysis.

### 2.5. Molecular Dynamics Simulation of SnocOBP9-Z3D Complexes

To clarify the interactions between SnocOBP9 and Z3D, we conducted time-evolution root-mean-square deviation (RMSD) analysis (Figure 7A). The SnocOBP9-Z3D complex was stable within 50 ns, with average values of about 3.3 Å (SD = 0.18 Å) for the complex, 2.63 Å (SD = 0.21 Å) for the protein, and 1.38 Å (SD = 0.19 Å) for the ligand. These results show that the 200 ns MD simulation was stable, suitable for use in subsequent analyses and satisfactory for determination of the main conformation of the SnocOBP9 complex. The results for the three MD simulation repetitions (Appendix A) were highly consistent and reliable. To further investigate the structure of the SnocOBP9-Z3D complex, the root mean square fluctuation (RMSF) (Figure 7B) was calculated when the SnocOBP9-Z3D conformation reached equilibrium (50~200 ns), and the local motion and flexibility were visualized. After the SnocOBP9-Z3D system reached equilibrium, the overall RMSF fluctuation level was low. The RMSF values of the selected amino acids were below 1 Å (MET54: 0.488 Å, PHE57: 0.689 Å, PHE71: 0.742 Å, PHE74: 0.655 Å, LEU116: 0.583 Å), further confirming the stability of the SnocOBP9-Z3D complex system. These key residues could interact with Z3D to form stable complexes in the dynamic simulation.

### 2.6. SnocOBP9-Z3D Binding Mode Analysis

The MD properties of the SnocOBP9-Z3D complex were further analyzed after MD runs through conformation sampling and clusters. We obtained 10 clusters in total, with cluster I and cluster II conformations comprising the highest proportions at 66% and 26.9% (Table 4). We superimposed 10 representative conformations of Z3D in the respective active site of SnocOBP9 (Figure 8). Z3D was always in the hydrophobic binding cavity of the protein, and its main interaction mode did not change during the simulation process.

As shown in Figure 9 and the 2D binding analysis (Appendix A), it can be predicted that the hydrophobic binding cavity was formed by several non-polar amino acid residues in the protein. MET 54, PHE 57, PHE 71, PHE 74 and LEU 116 contributed higher energy than the others. Among them, PHE71 and the hydroxyl hydrogen of Z3D have a long-distance hydrogen bond (3.8 Å), which may have the function to stabilize Z3D in the binding pocket.

### 2.7. SnocOBP9-Z3D Binding Energy Calculation

The binding free energy (Δ*G_bind_* −27.13 ± 0.043 kcal/mol) of SnocOBP9-Z3D was calculated and divided into four energy types (Van der Waal, electrostatic, polar and apolar solvation energy) (Table 5), representing the distinct free energy contributions to the binding affinity of the SnocOBP9 and Z3D complex. The contribution from Van der Waals energy (Δ*G_vdw_* −32.146 ± 0.045 kcal/mol) was significantly higher than those from other types of energy, while electrostatic energy (Δ*G_ele_* −4.595 ± 0.031 kcal/mol) and apolar solvation free energy (Δ*G_SA_* −3.163 ± 0.003 kcal/mol) contributions were fairly small (Table 5). Additionally, the polar solvation free energy (Δ*G_PB_* 12.773 ± 0.036 kcal/mol) was high, which implies that polar residues had an unfavorable effect on the binding affinity. Energy analysis showed that Δ*G_vdw_* was the main type of free binding energy and the binding affinity for SnocOBP9 and Z3D was mainly attributed to the hydrophobic interactions between them.

### 2.8. Per Residue Binding Free Energy Decomposition

The binding free energy contribution of each amino acid to the SnocOBP9-Z3D complex was further calculated with MM/PBSA. As shown in Figure 10A, MET54, PHE57, PHE71, PHE74 and LEU116 contributed more than −1 kcal/mol to the binding free energy of SnocOBP9-Z3D, with Δ*G_bind_* values of −1.153, −1.107, −1.303, −1.583 and −1.030 kcal/mol, respectively (Table 6). The centroid distance between the key amino acids with Z3D are shown in Figure 10B and Table 6.

The results indicate that the non-polar sidechains of these residues, which are essential for SnocOBP9 protein, participate in the formation of hydrophobic interactions with Z3D. Although the polar solvation energy for these key amino acids was unfavorable, the Δ*G_MM_* value of PHE74 was still as high as −2.094 kcal/mol, which contributed a large amount of the energy used for the binding process. Besides these key residues, other residues had little or no contribution to the binding affinity of SnocOBP9 to Z3D and had positive energy values, especially for LYS18 and ARG20. Therefore, we assumed that the nonpolar residues (MET54, PHE57, PHE71, PHE74 and LEU116) were crucial for the binding of SnocOBP9 with the ligand Z3D using hydrophobic interactions.

### 2.9. Computational Alanine Scanning

CAS analysis was used to evaluate the changes in the free binding energy after the key amino acids were substituted with alanine. We used the key residues of SnocOBP9 described above as the CAS targets to determine their importance in the binding process to Z3D.

Our CAS results showed that, in all tested residues of SnocOBP9 (MET54, PHE57, PHE71, PHE74, LEU116), substitution with alanine resulted in a mutation energy value ΔΔ*G_mut_* > 0.5 kcal/moL (Appendix A). According to the DS evaluation criteria, MET54, PHE57, PHE71, PHE74 and LEU116 may play important roles in stabilizing SnocOBP9-Z3D complex, consistent with the per-residue energy decomposition. Thus, alanine substitution at these key amino acids may damage the active conformation of SnocOBP9, resulting in a significant decrease in its binding affinity to insect pheromone Z3D.

### 2.10. Site-Directed Mutation and Binding Affinity

To confirm the results of CAS and clarify the key residues involved in the interactions between SnocOBP9 and Z3D, we conducted site-directed mutagenesis analysis. Five recombinant mutant proteins were prepared, and the Kd (K_1-NPN_) of mutant proteins with 1-NPN were 7.61 ± 0.92 μM (SnocOBP9M54A), 4.49 ± 0.82 μM (SnocOBP9F57A), 1.27 ± 2.42 μM (SnocOBP9F71A), 18.35 ± 1.66 μM (SnocOBP9F74A) and 17.51 ± 2.41 μM (SnocOBP9L116A) (Appendix A), respectively. The five mutant proteins were bound to Z3D, and the results show that the Ki value of SnocOBP9F57A binding with Z3D (Ki = 5.61 ± 0.50 μM) was significantly lower compared to that of wild-type SnocOBP9 (Figure 11). The fluorescence values of four proteins (MET54, PHE71, PHE74, LEU116) did not decrease below 50%, indicating that the mutant proteins had no binding affinity to Z3D; thus, the Ki value could not be calculated. These results are consistent with the per-residue binding free energy decomposition and CAS.

## 3. Discussion

The sequence analysis showed that SnocOBP9 had typical PBP structural characteristics. In our previous research on the olfactory system of *S. noctilio*, 16 OBPs were identified in *S. noctilio* via transcriptome analysis. Compared to the other 15 OBPs, SnocOBP9 is a homologue of pheromone-binding proteins and is closely-related to AmelASP1 in *A. mellifera* and MmedPBP1 in *Microplitis mediator* [56]. Based on sequence similarity and tissue specificity, AmelASP1 was assigned to the ASP1 family [58]. As a subclass of ASPs, AmelASP1 has been proposed to be associated with queen pheromone detection and as the opposite of ASP2 (interacting with other odorants) [59]. Meanwhile, studies have shown that release and storage of aggregated pheromones in *S. noctilio* may originate from the leg tendon gland of the hind legs, which is a centrally controlled storage. The leg tendon gland is a hollow reservoir starting in the femur and running down to the unguitractor plate, manubrium and arolium structures situated at the end of the clamp [48]. With a similar structure such as the tracer pheromone released by ant *Crematogaster*, which comes from the tibial tendon gland of their hind legs [60], females in *Ascogaster reticulatus* release (*Z*)-9-hexadecenal as a sex pheromone from the hind legs [61]; male *Polistes dominulus* are thought to release a lek-formation pheromone from their legs [62]; foragers in the stingless bee *Melipona seminigra* use a pheromone from the hind leg tendon gland to mark their food sites [63]. Further, *SnocOBP9* was present specifically in the antennae in *S. noctilio*, with no detectible expression in legs and a weak expression bias between the two sexes [56]. These results indicate that OBPs were primarily expressed in adult antennae and reflect their ability to enhance both the specificity and sensitivity of olfactory receptors to insect pheromones [64]. However, whether SnocOBP9 can bind to pheromones was unknown. Our study shows that Z3D could bind to SnocOBP9, and the fluorescence value could decrease to below 50%, resulting in the strongest binding affinity among the detected volatile compounds. This suggests that SnocOBP9 plays an essential role in the sexual behavior of wasps and participates in the pheromone transduction process.

Interestingly, SnocOBP9 could also bind with plant volatiles. The results show that the fluorescence value of (+)-Limonene and (−)-Limonene could also decrease to less than 50%. This observation highlights the flexibility of SnocOBP9 binding characteristics, although SnocOBP9 may be involved in sensing aggregate pheromones or sex pheromones. Similar results were obtained for the AcerOBP11 of *A. cerana*, which could bind with honeybee sex pheromones, such as queen mandibular pheromones (QMPs), methyl p-hydroxybenzoate (HOB), (E)-9-oxo-2-decanoic acid (9-ODA), alarm pheromone (n-hexanol), and plant volatiles such as 4-Allylveratrole [65]. AcerOBP15 in *A. cerana* has a strong binding affinity to most floral volatiles and a limited number of bee pheromone compounds, among which the compound with the highest affinity is myrcene [26]. In addition to OBPs, studies have shown that GOBPs and PBPs also possess binding flexibility. Traditionally, GOBPs are related to the recognition of host plant volatiles and PBPs are thought to engage in the recognition of insect pheromone volatiles [66]. However, a large number of studies have shown that the binding types for GOBPs and PBPs are not absolute. For example, the SexiGOBP2 of *Spodoptera exigua* displayed even stronger binding affinity to five sex pheromone components than to SexiPBP1 [67]. SlitGOBP1 and SlitGOBP2 in *S. litura* strongly bind to C14–C16 alcohol-pheromone analogs with high affinity [68]. CsupGOBPs in *Chilo suppressalis* can bind to not only host plant volatiles (farnesol, myrcene) but also sex pheromone components (Z13-18: Ald, Z11-16: Ald and Z9-16: Ald) [69]. Simultaneously, PBPs can bind to plant volatiles; for example, PxylPBPs in *Plutella xylostella* have weak binding affinities with six plant volatiles [70] and MsexPBP1 in *Manduca sexta* can bind some fatty acids [71]. Both LstiPBP1 and LstiPBP3 in *Loxostege sticticalis* were found to bind to host plant volatiles (α-lonone, β-lonone) [72]; PBP3 of *P. xylostella* had weak affinities for all tested plant volatiles [73]. As for *S. noctilio*, before our fluorescence binding assay experiment, outdoor experiments had already demonstrated that using only insect pheromones as a trap had limited effectiveness in the forest [74]. Interestingly, the use of transparent baffle traps containing pheromones and host plant volatiles (including (+)-Limonene and (−)-Limonene) was more effective [75]. Therefore, we hypothesized that SnocOBP9 may be involved not only in female recognition of the pheromones released by adult males, but also in male and female recognition of volatiles derived from host trees, where both sexes meet and mate. It is worth noting that most insect OBPs fluorescence competition experiments use 1-NPN as the fluorescent probe, so we did not conduct quenching experiments and molecular dynamics experiments on 1-NPN. Therefore, the unresolved issue of the binding site of SnocOBP9 to 1-NPN and the mechanism of fluorescence decrease needed to be solved in future experiments.

The binding energy of the SnocOBP9 and Z3D complex was subsequently calculated, and the results show that Van der Waals energy was the dominant binding energy and that it played an important role in the interactions between the pheromone and protein (Δ*G_vdw_* −32.146 ± 0.046 kcal/mol and Δ*G_ele_* −4.600 ± 0.031 kcal/mol). Similar studies have also been reported. For instance, in the course of CpomPBP1-codlemone complex formation in *Cydia pomonella*, ∆*G_vdw_* (−33.09 kcal/mol) was the main driving force and determined the affinity of codlemone to the binding pocket of CpomPBP1 [76]. In *Athetis lepigone*, Van der Waals energy (−42.235 ± 0.088 kcal/mol for AlepPBP1-Z7-12: Ac and −48.315 ± 0.118 kcal/mol for AlepPBP1-Z9-14: Ac) was significantly higher than other types of energy, whereas the electrostatic and apolar solvation free energy contributions were fairly low [77]. However, in addition to Van der Waals energy, electrostatic energy could also be a dominant binding energy. In the simulation of laccase secreted by an *S. noctilio* symbiotic fungus, *A. areolatum*, with lignin ligands, 30 ns MD analysis revealed that the binding energies of laccase AaLac2, AaLac3 and AaLac8 to the lignin ligand were low, and that there was a large number of hydrogen bonds between the laccase and lignin ligand through molecular docking [78].

Per-residue free energy decomposition and CAS are useful for evaluating the key amino acid residues of protein–ligand complexes [79]. Per-residue free energy decomposition results showed that Δ*G_bind_* of the five non-polar amino acids, MET54, PHE57, PHE71, PHE74 and LEU116, was higher than −1 kcal/mol, and the Δ*G_MM_* of PHE74 was as high as −2.094 kcal/mol. These amino acids generally had low RMSF values, indicating that the amino acid residues in the active site region of SnocOBP9 were relatively stable [80]. However, although the 2D diagram plot and per-residue free energy decomposition results show that the three amino acids MET54, PHE57 and LEU116 all contributed similar energy to Z3D, the distance between amino acid LEU116 and Z3D is farther. This may be related to the template we selected (3cdn_A). Marion E. Pesenti et al. suggested that the binding site integrity of AmelASP1 depends on C terminus (111–119) conformation. According to our protein sequences alignment result (Figure 6B), 112–119 has relatively high conservation and belongs to active sites, and LEU116 is located in it. Meanwhile, leucine belongs to a class of amino acids with longer side chains, so the replacement of LEU116 with alanine may have a comparatively large impact on the binding ability of the protein [81]. The per-residue free energy decomposition and CAS methods were also used for the *A. lepigone*, AlepPBP2-Phoxim (PHE35, PHE39, ILE55, ILE65, ILE97 and PHE122) and AlepPBP3-Phoxim (PHE12, PHE52 and ILE134) complexes. The Δ*G_bind_* of key residues was higher than −1 kcal/mol, implying that these key residues could interact with phoxim to form stable complexes [82]. In the complex of GmolPBP2-Z8−12: Ac (*Grapholita molesta*), the per-residue free energy decomposition was higher than −2 kcal/mol for PHE11 and ILE51 and higher than −1 kcal/mol for CYS32, PHE35, ILE93 and PHE116, suggesting that these residues were involved in the formation of the hydrophobic pocket where Z8−12: Ac binds [83]. These findings imply that the key amino acid residues were involved in hydrophobic interactions with ligands and also provided the dominant driving force for the formation of the binding affinity in protein–ligand complexes. The free energy changes after amino acid substitution with an alanine residue were mainly evaluated using CAS [84], and our CAS data also indicate that ΔΔ*G_mut_* for these five key residues was higher than 0.5.

CAS analysis is an excellent method to obtain key residues but cannot predict the key residues with 100% accuracy. Consequently, we employed a reverse chemical ecology approach by mutating the amino acids of OBPs to confirm the key residues of proteins and further verify the protein function [31]. After replacing the three lysine residues (Lys123) at the C-terminal of HarmOBP7 (*Helicoverpa armigera*) with methionine, the results showed that the affinity for the two pheromone components decreased, thus indicating that Lys123 was involved in the formation of functional groups in the protein–ligand complex [85]. In the larvae of *P. xylostella*, five aromatic residues of GOBP2 in the binding pocket were replaced with leucine, and the results show that only one or two amino acid substitutions could completely abolish the binding ability to pheromone, confirming that PxylGOBP2 is a narrowly tuned binding protein [86]. Therefore, we conducted a site-directed mutagenesis assay for SnocOBP9WT. The fluorescence competitive assay values for four variants (MET54, PHE71, PHE 74 and LEU116) and Z3D never decreased below 50%, indicating their significance in the formation of the SnocOBP9-Z3D complexes. This result is consistent with previous computational simulation results and suggests that the binding residues of insect OBPs usually participate in the formation of hydrophobic pockets, which use Van der Waals energy and hydrogen bonds to bind different ligands [87,88]. However, the fluorescence value for SnocOBP9F57A-Z3D was less than 50% (Ki = 5.61 ± 0.50 μM). Similar results have been recently reported. For AlepPBP2 and AlepPBP3 in *A. lepigone*, AlepPBP2L55A, AlepPBP2L65A, AlepPBP2F122A and AlepPBP3L52A showed no significant differences with AlepPBP2WT (wild-type AlepPBP1) and AlepPBP3WT (wild-type AlepPBP2) in binding with phoxim [80]. For CpomPBP1 in the codling moth *C. pomonella*, the three mutated proteins of CpomPBP1 (CpomPBP1F36A, CpomPBP1I52A, CpomPBP1I94A) had no remarkable negative effects on their ability to bind to codlemone [89]. Therefore, CAS should not be used as the sole method for calculating or predicting key residues. Other methods such as site-directed mutation or binding affinity should be further conducted to confirm the reliability of the CAS results.

In conclusion, (1) we first determined that SnocOBP9 has a high affinity with Z3D, and revealed the interaction between SnocOBP9 and Z3D by computational simulation and fluorescence binding assay experiments after site-directed mutagenesis of five hydrophobic residues (MET54, PHE57, PHE71, PHE74 and LEU116). Our results regarding the interactions between SnocOBP9 and Z3D will contribute to the design and discovery of novel chemicals with similar functions. (2) In addition, some putative or suspected components, including (*Z*)-3-octen-ol, (*Z*)-3-dodecen-ol, (*Z*)-4-decen-ol and (*E, E*)-2,4-decadienal, may play important roles in these interactions. The mechanisms underlying their interactions with other proteins can be investigated through computer simulation and molecular experiments before application in the field. (3) These findings have powerful theoretical and practical significance for attractive drug design or chemical ecological prevention and control, and will help to develop target trap lures through computational simulation design.

## 4. Materials and Methods

### 4.1. Insect Rearing and Tissue Collection

In late May 2019, five trees with *S. noctilio* hazard characteristics were collected at the Xindian Forest Station in Duerbote Mongolian Autonomous County, Daqing City, Heilongjiang Province, China (N 46°37′47′′, E 124°25′51′′), and sent to the Plant Quarantine Laboratory of Beijing Forestry University. Adults were collected every day, male adults were dissected (unmated), immediately frozen in liquid nitrogen and stored at −80 °C for later use.

### 4.2. Total RNA Isolation, cDNA Synthesis and Sequence Analysis

Total RNA was extracted from the antennae of 10 males using the RNeasy Plus Mini Kit (No. 74134; QIAGEN, Hilden, Germany) following the manufacturer’s instruction, and this experiment was repeated 3 times. RNA quality and concentration were measured using the ultramicro-spectrophotometer NanoDrop 8000 (Thermo, Waltham, MA, USA). The first-strand complementary DNA (cDNA) was synthesized using RNA (1 μg) and the PrimeScript RT Reagent Kit with the gDNA Eraser Kit (No. RR047A; TaKaRa, Shiga, Japan) following the instructions.

Analysis of the antennal transcriptome of *S. noctilio* indicated that the SnocOBP9 gene has an open reading frame (ORF) of 426 bp that includes a sequence encoding a signal peptide (63 bp), which indicates it is a complete gene. The amino acid sequence was aligned and analyzed using WebLogo 3 Server (https://weblogo.berkeley.edu/logo.cgi, accessed on 20 August 2021) and ClustalX 2.0 [90].

### 4.3. Cloning, Expression and Purification of Recombinant Wild-Type SnocOBP9 Protein

Part of the coding region (ORF) of SnocOBP9, with the sequence encoding the signal peptide omitted, was amplified with PCR using gene-specific primers containing the restriction enzyme sites for *EcoRI* in the forward primer (SnocOBP9WT-F: 5′-CCGGAATCCAAGCTTCCCG ATTGGGTACC-3′) and *XhoI* in the reverse primer (5′-CCGCTCGAGTTACAACACGTACCATAATT CTGGAGC-3′) (Table 7). cDNA from the antenna was used as the template. The amplification conditions were 34 cycles of 98 °C for 10 s, 55 °C for 50 s, and 72 °C for 5 s, and the process was performed using Ex Taq DNA polymerase (No. RR001C; Takara, Dalian, China). After analysis on 1.5% agarose gel, the PCR product was purified with the Axygen Gel Extraction Kit (No. AP-GX-250; Axygen, NY, USA) and cloned into the plasmid pEASY-T1 (No. CT101-01; Transgen, Beijing, China). The plasmids were transferred into *Escherichia coli* (*E. coli*) pEASY-Blunt3 competent cells (No. CB301-01; Transgen, Beijing, China). Positive clones were selected by PCR and sequenced (RuiBio Biotech, Beijing, China).

Plasmids were extracted with the Axyprep Plasmid Miniprep Kit (No. AP-MN-P-250; Axygen, NY, USA), and digested with FlyCut *EcoRI* (No. JE201-01; Transgen, Beijing, China) and FlyCut *XhoI* (No. JX201-01; Transgen, Beijing, China). The fragment encoding the correct SnocOBP9 sequence was purified and sub-cloned into the bacterial expression vector pET-28a (+) (No. P3110; Solarbio, Beijing, China). Afterward, the plasmids containing the correct insert (pET-28a (+)-SnocOBP9) were transformed into *E. coli* BL21 (DE3) pLysS chemically competent cells (No. CD701-02; Transgen, Beijing, China). Expression of SnocOBP9 was induced with isopropyl-β-D-thiogalactopyranoside (IPTG) (No. GF101-01; Transgen, Beijing, China) at a final concentration of 1 mM at 37 °C for 6 h.

Samples were separated using an ultrasonic breaker on ice for 12 min and centrifuged at 4 °C and 12,000 rpm for 15 min to completely separate the supernatant and the competent cell fragments. The supernatant and pellet were then analyzed with sodium dodecyl sulfate polyacrylamide-gel electrophoresis (SDS-PAGE). SnocOBP9 was observed as inclusion bodies in the pellet, which were purified with RoteinIso Ni-NTA Resin (No. DP101-01; TransGen, Beijing, China). We obtained a protein solution by denaturing the inclusion bodies, followed by renaturation with 8-0.5 M urea renaturation buffer. The renaturation process is detailed in the Appendix A. The His-tag was removed with the Enterokinase excision kit (No. NRPB14S; Nuptec, Hangzhou, China). The protein was concentrated using Amicon Ultra concentrators with a 10 kDa cutoff (Millipore, Billerica, MA, USA), and purity was confirmed by SDS-PAGE analysis. The concentration of SnocOBP9 protein was quantified using the Bradford method with bovine serum albumin (BSA) as the standard protein.

### 4.4. Identification of Recombinant SnocOBP9 (Western Blot, LC-MS/MS and CD-Spectrum)

After transfer membrane electrophoresis, the surface of PVDF membranes was washed twice with PBST for 5 min. Then, we took 0.5 g of skim milk (5%) powder, fully dissolved in 10mL of PBST, to make the liquid level slightly higher than the membrane surface for sealing. After completing the blocking, we removed the skim milk powder and washed the PVDF membrane with PBST three times for 5 min each time. The 6*His-Tag MouseMonclonal antibody was added, (No. D191001, Sangon Biotech, Shanghai, China)) diluted with skim milk powder, and placed on a horizontal shaker for incubation at 4 °C overnight. The liquid was discarded and the PVDF membrane was washed 3 times with PBST for 5 min each time after completing the primary antibody incubation. The Goat Anti-Mouse IgG H&L (HRP) was added, preadsorbed (No. ab97040, Abcam, Shanghai, China), diluted with skim milk powder, then placed on a horizontal shaker for incubation at room temperature for 2 h. Finally, after secondary antibody incubation, the PVDF membrane was washed with PBST three times, then color development and fixation were finished.

The chromatographic fraction was analyzed by Orbitrap Fusion Lumos LC-MS/MS (Thermo Fisher Scientific, Waltham, MA, USA). Briefly, the proteins were in-solution-digested by 100 ng/mL trypsin, the peptides were extracted by a mixture of formic acid/acetonitrile/distilled water, then separated using Acclaim PepMap RSLC C18 column (Thermo Fisher Scientific, Waltham, MA, USA). The separated peptides were analyzed by LC-MS/MS. The raw MS data was analyzed and searched against target protein database based on the species of the samples using Maxquant (1.6.2.10). The parameters were set as follows: the protein modifications were carbamidomethylation (C) (fixed), oxidation (M) (variable), Acetyl (Protein N-term) (variable); the enzyme specificity was set to trypsin; the maximum for missed cleavages was set to 2; the precursor ion mass tolerance was set to 20 ppm, and MS/MS tolerance was 20 ppm. Only high-confident identified peptides were chosen for downstream protein identification analysis.

CD spectra of SnocOBP9 were obtained and used to investigate the protein secondary structure by using Circular dichroism (CD) (MOS-500) (Biologic, Beijing, China) with a 1 nm Cuvette. Before measurement, the SnocOBP9 protein was dialyzed by ddH_2_O, and the spectra were recorded from 180 to 260 nm, acquisition time 1 s/point, cuvette width 0.1 cm. Finally, the acquired data were analyzed using the CDNN program (http://gerald-boehm.de, accessed on 17 June 2021).

### 4.5. Fluorescence Competitive Binding Assay

To measure the binding affinity of the ligand to recombinant SnocOBP9, 1-NPN was selected as a fluorescent probe. Eleven host volatiles ((−)-Limonene, (+)-Limonene, 3-Carene, Alpha-Pinene and Beta-Pinene, Camphene, Beta-Myrcene, Beta-Ocimene, Sabinene, Hexanal and Eucalyptol), three insect pheromones ((*Z*)-3-Decen-ol, (*Z*)-4-Decen-ol and (*E, E*)-2,4-decadienal) and six symbiotic fungal volatile (Cyclopentanone, Geraniol 2-Hexene, (−)-Globulol, Linalol and 1-(3-ethylphenyl)ethenone) compounds were selected to bind with recombinant SnocOBP9. A multi-scan Spectrum Molecular Device Spectra Max i3 Fluorescence Spectrophotometer (Thermo Scientific, Wilmington, DE, USA) was used to conduct fluorescence competitive binding assays with an excitation wavelength of 337 nm. The emission spectra were recorded between 390 and 500 nm. The parameters were selected such that the slit widths for both excitation and emission were 10 nm. High-speed scanning was used for spectra recording. SnocOBP9 was prepared as a 2 µM solution in 20 mM Tris-HCL buffer (pH 7.4), and the ligands were dissolved in chromatographically pure methanol as 1 mM stock solutions. The affinity of SnocOBP9 for the labeled probe was determined by adding aliquots of 1-NPN stock solution to prepare concentrations ranging from 2 µM to 20 µM. The binding affinity of SnocOBP9 for the different ligands was estimated using competitive binding assays with both 1-NPN and SnocOBP9 at 2 µM. The final concentrations of the 12 competitive ligands and analog compounds were in the range of 2–20 µM. SnocOBP9 binding with every component was replicated three times. Intensity values corresponding to the maximum fluorescence emission were plotted against the free ligand concentrations to calculate the dissociation constant. Assuming the activity of the protein was 100%, and the stoichiometric ratio between the protein and the ligand was 1:1 when saturated, the bound ligand was determined from the fluorescence intensity value. The K_1__-NPN_ value was calculated by linearizing the curve using Scatchard plots. Finally, the following equation was used to calculate the dissociation constants of the competitors (Ki):Kd = [IC50]/(1 + [1-NPN]/K_1-NPN_)(1)
where [IC50] is the concentration of the competitor that halves the initial fluorescence intensity, [1-NPN] represents the free concentration of 1-NPN and K_1-NPN_ represents the dissociation constant between SnocOBP9 and 1-NPN.

### 4.6. Homology Modeling and Molecular Docking

*A. mellifera* antennal-specific protein 1 (ASP1; PDB ID: 3cdn_A) was selected as the optimal template for constructing the SnocOBP9 model using Modeller 9.25 [91,92,93,94]. SAVES v5.0 (https://servicesn.mbi.ucla.edu/SAVES/, accessed on 22 August 2020) was used to confirm the residue compatibilities and stereochemical rationalities of SnocOBP9. The protein–ligand binding mode of SnocOBP9 was obtained with Autodock Vina 1.1.2 molecular docking, and the docking input files were acquired with Autodock Tools 1.5.6 [95].

Autogrid boxes (binding pockets) were selected using Protein plus DoGSiteScore (https://protein.plus/, accessed on 22 August 2020). The parameters were used as described in the Autodock Vina manual. Ten poses for each ligand were obtained. The docking scores for the protein–ligand complexes were judged using Vina and visual structure analysis was performed with VMD 1.9.3 and PyMOL 2.4.1 [96,97].

### 4.7. Molecular Dynamics Simulations

The GROMACS2019.6 software package was used to conduct MD simulations for the SnocOBP9-Z3D complex [98]. The parameters and charges of Z3D were obtained using the AmberTools 18 GAFF force field, which was optimized with the ACPYPE script [99,100]. The SnocOBP9 protein was optimized with the AMBER-ff99sb-ildn force field by GROMACS software [101]. The SnocOBP9-Z3D complex was solvated in a cubic box with a length of 8 Å filled with 6215 TIP3P water molecules. The total charge of the SnocOBP9-Z3D system was −12; therefore, 12 Na+ ions were added to neutralize the charge. The system was minimized with a conjugated gradient (CG) to remove incorrect contacts. Position-restricted MD simulations were performed to relax water solvents using the constant number, volume and temperature (NVT) and the constant number, pressure and temperature (NPT) ensembles. MD runs were simulated for 200 ns with a time step of 2 fs, and temperature and pressure were coupled at 300 K and 1 bar using the V-rescale thermostat and Parrinello–Rahman barostats, respectively. Long-range electrostatic interactions were calculated using the Particle Mesh Ewald (PME) algorithm. The LINear Constraint Solver (LINCS) algorithm was used to constrain all covalent bonds with hydrogen atoms. The cut-off radii for both Coulomb and Van der Waals interactions were set to 10 Å. Coordinates were recorded every 2 ps, resulting in 100,000 uncorrelated configurations. The equilibrium of the SnocOBP9-Z3D complex was determined by analyzing the root mean square deviation (RMSD) of the complex, protein and ligand. Gromacs cluster was used to identify similar structures, and the representative dominant conformations were defined as the structures at the center of the largest cluster.

### 4.8. Binding Free Energy Calculation

Complexes in equilibrium conformations were selected and used to calculate the binding free energy (Δ*G_bind_*). The molecular mechanics/Poisson–Boltzmann surface area (MM/PBSA) method and g_mmpbsa tool were used to calculate (Δ*G_bind_*) [102,103], which was defined as:Δ*G_bind_* = Δ*E_MM_* + Δ*G_PB_* + Δ*G_SA_* − *T*Δ*S*(2)
where ΔE_MM_ is the potential electrostatic and Van der Waals energy, and Δ*G_PB_* and Δ*G_SA_* are the solvation free energies of the polar and nonpolar solvents. As the contributions of −*T*Δ*S* are similar in the same types of protein systems, and the computational cost is high, the parameter can be neglected [104].

To identify the key amino acid residues in the binding interactions between SnocOBP9 and the chemical ligand Z3D, the binding free energy of each residue was decomposed into Van der Waals energy, electrostatic energy, polar solvation free energy and non-polar solvation free energy using the MM/PBSA method [102]. Residues that contributed binding energies greater than −1 kcal/mol were considered the key residues in the binding affinity between SnocOBP9 and Z3D [83].

### 4.9. Computational Alanine Scanning and Site-Directed Mutation

CAS is considered a reliable tool for determining key residues in protein–protein and protein–ligand interactions [103,104]. The Discovery Studio (DS) protocol is regarded as a reliable tool and has been tested on large sets of mutation experiments. Following previous studies, the Calculate Mutation Energy (Binding) module in the Discovery Studio 4.5 (Accelrys Inc., San Diego, CA, USA) platform was employed to measure the binding free energy changes (Mutation Energy, ΔΔ*G_mut_*) before and after amino acid (alanine) substitutions [105],
∆∆*G_mut_* = ∆*G_bind_* (mutant) − ∆*G_bind_* (wild-type)(3)
where ∆*G_bind_* (mutant) and ∆*G_bind_* (wild-type) are the binding free energies in mutated and wild-type complexes, and ∆∆*G_mut_* is the binding free energy difference between wild-type and mutants. ∆∆*G_mut_* values between −0.5 and 0.5 indicate the mutant residues had no significant effect on affinity; ∆∆*G_mut_* values above 0.5 indicate the mutant residues induced weaker interactions and a decrease in affinity; ∆∆*G_mut_* values below −0.5 indicate the mutant residues enhanced the binding affinity for protein–ligand interactions [106].

Mutant amino acids (MET54, PHE57, PHE71, PHE74, LEU116) were determined from the CAS results, and the primers of mutant proteins were designed with NCBI Primer-Blast (Table 7). A total of 50 microliters of SnocOBP9 expression-competent *E. coli* were inoculated into 100 mL LB medium and cultured overnight. The plasmid was extracted with the EasyPure HiPure Plasmid MiniPrep Kit (No. EM101-01; TransGen, Beijing, China) and used as the PCR template. The PCR assay for the site-directed mutation of SnocOBP9 was performed following the instructions for the Fast Mutagenesis System kit (No. FM111-01, TransGen, Beijing, China) with 30 cycles of 95 °C for 30 s, 62 °C for 50 s and 76 °C for 30 s. The reaction products were transformed to pEASY-Blunt3 competent cells. Plates were seeded with the cells, and monoclonal colonies were selected. The mutant SnocOBP9 genes were verified by gene sequencing.

The expression and purification processes for mutant SnocOBP9 (SnocOBP9MT: SnocOBP9M54A, SnocOBP9F57A, SnocOBP9F71A, SnocOBP9F74A and SnocOBP9L116A) proteins were similar to those for SnocOBP9WT. SnocOBP9MT proteins were used for the competitive binding assay with Z3D.

### 4.10. Statistical Analysis

Data (mean ± SE) from various samples were analyzed with one-way analysis of variance (ANOVA) using SPSS Statistics version 21.0, followed by Tukey’s multiple comparison test for mean comparison.

## Figures and Tables

**Figure 1 ijms-23-08456-f001:**
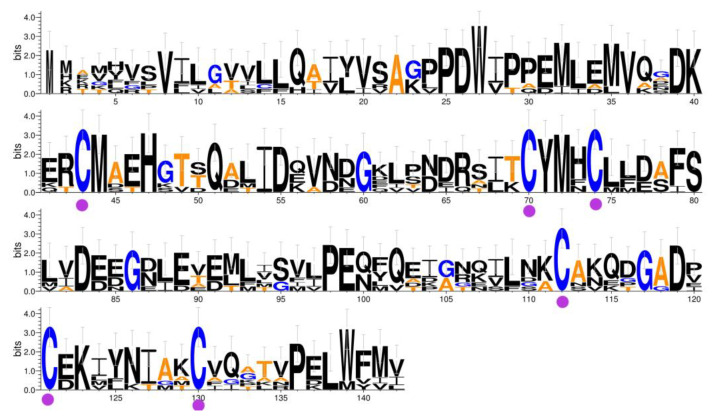
The sequences of SnocOBP9 in *S. noctilio* and the other six PBPs of Hymenoptera were aligned. Sequence conservation was determined based on the overall height of the stack at that position, and the height of the symbols within the stack suggests the relative frequency of each amino acid at that position. Conserved cysteines are marked with purple dots. The insect species and GenBank accession numbers are the following: *Formica exsecta* PBP (XP_029680467.1), *Belonocnema treatae* PBP (XP_033213373.1), *Nylanderia fulva* PBP (XP_029165894.1), *Orussus abietinus* PBP (XP_012277309.1), *Dinoponera quadriceps* PBP (XP_014476791.1), *Odontomachus brunneus* PBP (XP_032663926.1).

**Figure 2 ijms-23-08456-f002:**
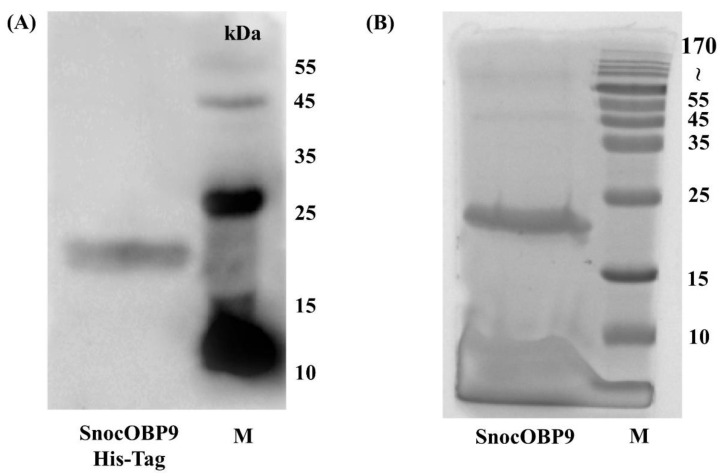
(**A**) Western blot analysis of SnocOBP9 with 6*His-Tag. Lane M represents molecular marker. (**B**) SDS-PAGE analyses of the expression and purification of recombinant SnocOBP9. Lane SnocOBP9 represents purified recombinant protein; lane M represents molecular marker.

**Figure 3 ijms-23-08456-f003:**
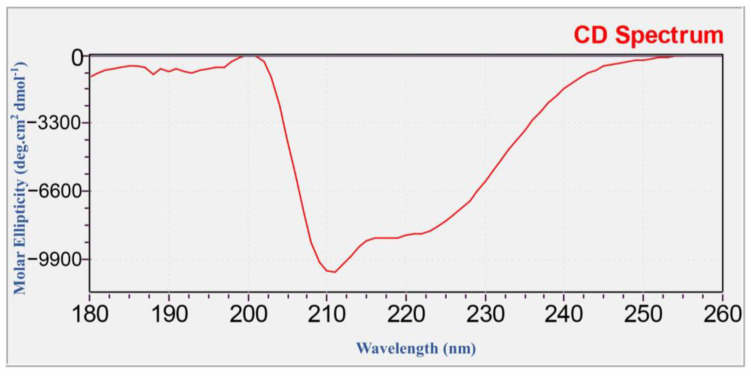
UV (180–260 nm) scan of SnocOBP9. The abscissa represents the scanning wavelength, and the ordinate represents the molar ellipticity [θ] (deg·cm^2^ dmol^−1^).

**Figure 4 ijms-23-08456-f004:**
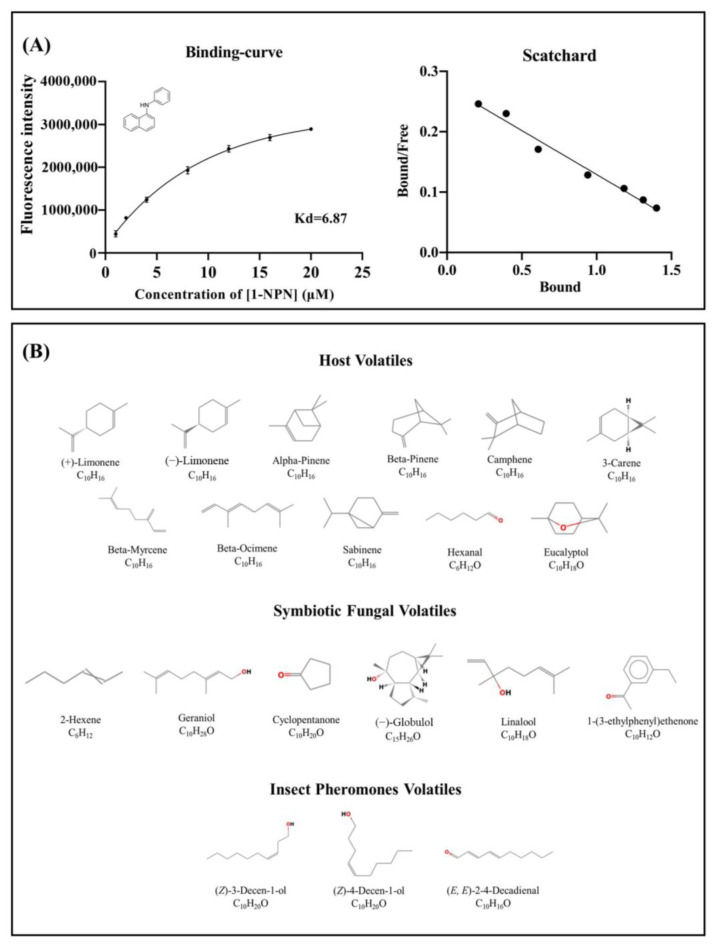
(**A**) Binding curve for 1-NPN to SnocOBP9 with Scatchard plot. Dissociation constant was Kd = 6.87 ± 0.73 μM. (**B**) The chemical structure.

**Figure 5 ijms-23-08456-f005:**
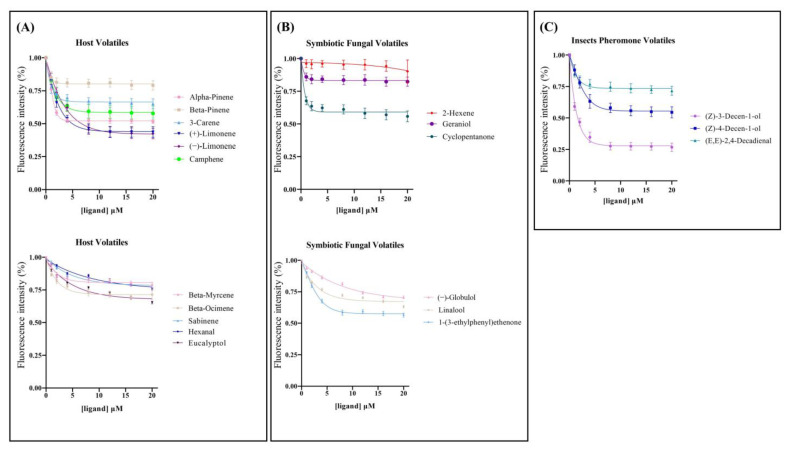
Binding curves for host volatiles (**A**), symbiotic fungal volatiles (**B**) and insect pheromone volatiles (**C**) to SnocOBP9. The ligand names are shown on the right side in the curves. The binding data are listed in Table 3.

**Figure 6 ijms-23-08456-f006:**
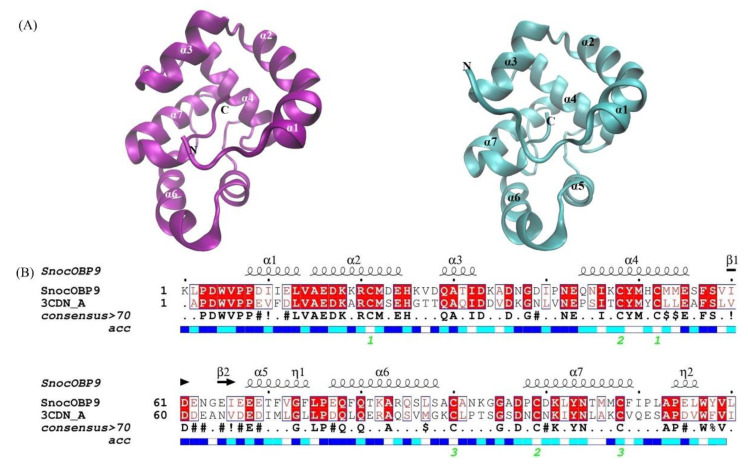
(**A**) Structural modeling of SnocOBP9; templates *A. mellifera* ASP1 (PDB ID: 3cdn_A) and model SnocOBP9. (**B**) The amino acid sequences alignment of SnocOBP9 and the template *A. mellifera* pheromone-binding protein (AmelASP1); 3CDN_A represents AmelASP1. Disulfide bridges rendered in green digits “1−3”.

**Figure 7 ijms-23-08456-f007:**
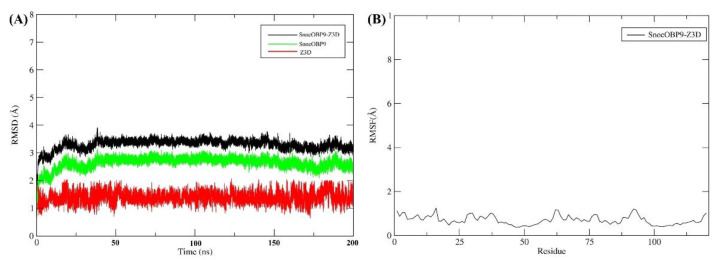
(**A**) The time-evolution RMSD curves (200 ns) of the SnocOBP9-Z3D complex, SnocOBP9 and Z3D. (**B**) RMSF curve of SnocOBP9-Z3D complex (50~200 ns).

**Figure 8 ijms-23-08456-f008:**
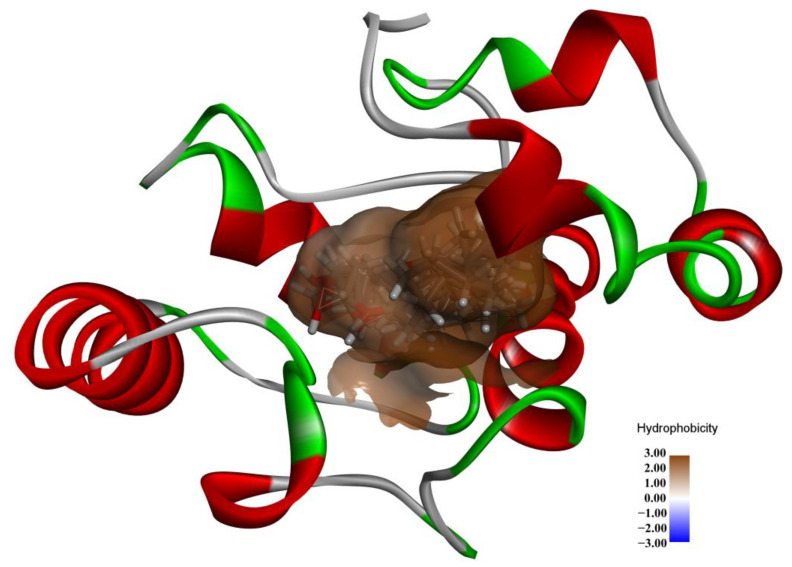
The hydrophobic cavities of Z3D (ten representations) anchored in SnocOBP9 displayed in surface view.

**Figure 9 ijms-23-08456-f009:**
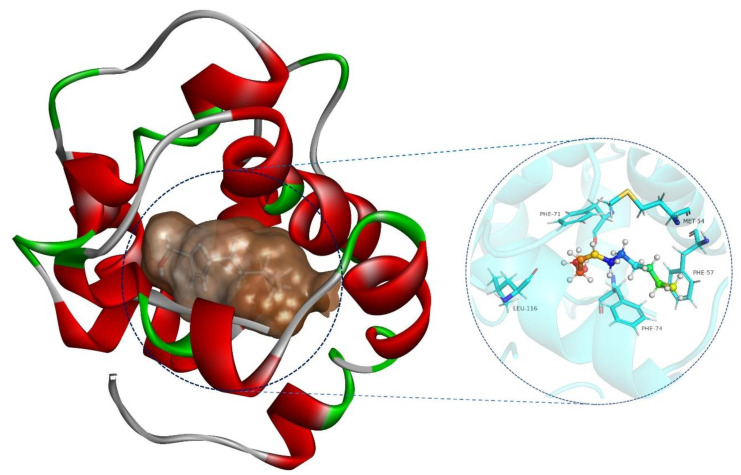
Representative conformation (Cluster I) for the SnocOBP9-Z3D complex produced based on the MD simulation trajectories. Representative residues, including MET54, PHE57, PHE71, PHE74 and LEU116, are marked on the binding interface.

**Figure 10 ijms-23-08456-f010:**
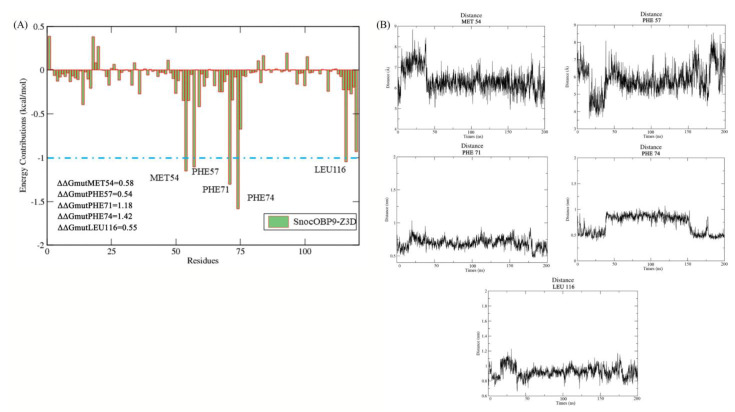
(**A**) Each residue for the SnocOBP9-Z3D complex calculated from the equilibrated conformations with 100–200 ns MD. Residues contributing more than −1.00 kcal/mol to the binding free energy are marked by the blue line. (**B**) Centroid distance between the five key amino acids and Z3D.

**Figure 11 ijms-23-08456-f011:**
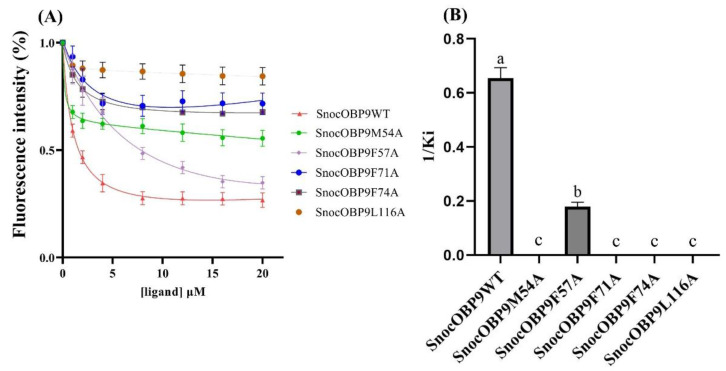
(**A**) Binding curve of SnocOBP9WT and mutant SnocOBP9 proteins. SnocOBP9WT represents the wild-type SnocOBP9 proteins. SnocOBP9M54A, SnocOBP9F57A, SnocOBP9F71A, SnocOBP9F74A and SnocOBP9L116A refer to MET54, PHE57, PHE71, PHE74 and LEU116 substitutions with ALA in SnocOBP9. (**B**) Comparison of the binding affinities (indicated by 1/Ki) for wild-type (WT) SnocOBP9 and SnocOBP9F57A to Z3D. Different letters indicate significant differences between SnocOBP9WT and SnocOBP9F57A (*p* < 0.05, ANOVA, Tukey’s HSD).

**Table 1 ijms-23-08456-t001:** The LC-MS/MS analysis of SnocOBP9.

Protein ID	Majority Protein ID	Sequence Coverage	Unique Sequence Coverage	Unique Peptides	Weight ^a^	Score ^b^	Intensity ^c^
SnocOBP9	SnocOBP9; A0A857N3E7	19.1%	19.1%	3	16.162	323.31	3.5801 × 10^9^

^a^ All weight values are given in “bp”. ^b^ The protein credibility score, the credibility of the protein increases with the value. ^c^ Relative abundance of proteins.

**Table 2 ijms-23-08456-t002:** The LC-MS/MS analysis of the three peptides in SnocOBP9.

Peptides Sequence	Length ^a^	Mass ^c^	Proteins	Start Position ^a^	End Position ^a^	Score ^b^
ADNGDIPNEQNIK	13	1426.6688	SnocOBP9;A0A857N3E7	56	68	112.93
CMDEHKVDQATIDKADNGDIPNEQNIK	27	3097.4081	SnocOBP9;A0A857N3E7	42	68	118.41
VDQATIDKADNGDIPNEQNIK	21	2297.1135	SnocOBP9;A0A857N3E7	48	68	107.33

^a^ All length values are given in “bp”. ^b^ The peptide credibility score, the credibility of the protein increases with the value. ^c^ Molecular weight of the peptide, values are given in “Dalton”.

**Table 3 ijms-23-08456-t003:** Binding data for different types of volatiles to SnocOBP9.

Ligand Name	CAS Number	IC_50_ ^a^	Ki ^a^
**Host Volatiles**			
(+)-Limonene	5989-27-5	6.09 ± 1.43	4.87 ± 0.94
(−)-Limonene	5989-54-8	6.95 ± 0.61	5.50 ± 0.49
Alpha-Pinene	80-56-8	>20 ^b^	— ^c^
Beta-Pinene	127-91-3	>20	—
3-Carene	13466-78-9	>20	—
Camphene	79-92-5	>20	—
Beta-Myrcene	123-35-3	>20	—
Beta-Ocimene	13877-91-3	>20	—
Sabinene	3387-41-5	>20	—
Hexanal	66-25-1	>20	—
Eucalyptol	470-82-6	>20	—
**Symbiotic Fungal Volatiles**			
Geraniol	106-24-1	>20	—
2-Hexene	592-43-8	>20	—
Cyclopentanone	120-92-3	>20	—
(−)-Globulol	489-41-8	>20	—
Linalool	78-70-6	>20	—
1-(3-ethylphenyl)ethenone	22699-70-3	>20	—
**Insect Pheromones Volatiles**			
(*Z*)-3-Decen-ol	10340-22-4	1.92 ± 0.14	1.53 ± 0.09
(*Z*)-4-Decen-ol	57074-37-0	>20	—
(*E*, *E*)-2,4-Decadienal	25152-84-5	>20	—

The protein and 1-NPN in the mixtures were both at concentrations of 2 μM. The mixture was titrated with 0.1 mM solutions per ligand to a final concentration of 1–20 μM. ^a^ All concentration values are given in μM; ^b^ IC_50_ > 20 means that fluorescence value cannot decrease below 50%; ^c^ Ki could not be calculated as “—”.

**Table 4 ijms-23-08456-t004:** Cluster analysis of the SnocOBP9-Z3D complex based on the trajectory of MD simulations.

System	Clusters	Occurrence [%]
SnocOBP9-Z3D	I	66.02
	II	26.86
	III	2.09
	IV	1.52
	V	0.95
	VI	0.88
	VII	0.67
	VIII	0.50
	IX	0.31
	X	0.19

**Table 5 ijms-23-08456-t005:** The binding free energy of the SnocOBP9-Z3D complex.

Complex	Sampling Conformations	Cluster ^b^	Van Der Waal Energy ^a^ (Δ*G_vdw_*)	Electrostatic Energy ^a^ (Δ*G_ele_*)	Polar Solvation Energy ^a^ (Δ*G_PB_*)	Apolar Solvation Energy ^a^ (Δ*G_SA_*)	Total Energy ^a^ (Δ*G_bind_*)
SnocOBP9-Z3D	50,000	100–200	32.146 ± 0.045	−4.595 ± 0.031	12.773 ± 0.036	−3.163 ± 0.003	−27.13 ± 0.043

^a^ All energy values are given in kcal/mol; ^b^ Cluster time value is given in “ns”.

**Table 6 ijms-23-08456-t006:** The decomposition for the important residues contributing to the binding free energy and the average centroid distance between the five key amino acids to Z3D.

Complex	Residue	MM Energy ^a^ (Δ*G_MM_*)	Polar Energy ^a^ (Δ*G_PB_*)	Apolar Energy ^a^ (Δ*G_SA_*)	Total Energy ^a^ (Δ*G_bind_*)	Average Distance ^b^	Standard Deviation ^b^
SnocOBP9-Z3D	MET-54	−1.412	0.332	−0.073	−1.153	6.3592	0.5401
	PHE-57	−1.328	0.318	−0.097	−1.107	5.8364	0.8025
	PHE-71	−1.776	0.582	−0.109	−1.303	6.9383	0.7486
	PHE-74	−2.094	0.644	−0.134	−1.583	7.1332	1.7607
	LEU-116	−1.326	0.329	−0.033	−1.030	9.2189	0.7678

^a^ All energy values are given in kcal/mol. ^b^ All distance values are given in Å.

**Table 7 ijms-23-08456-t007:** Primers for protein expression.

Primer Name	Primer Sequence (5′-3′)
SnocOBP9WT-F	CCGGAATCCAAGCTTCCCG ATTGGGTACC
SnocOBP9WT-R	CCGCTCGAGTTACAACACG TACCATAATT CTGGAGC
SnocOBP9M54A-F	GTTACATGCATTGCATGGCGGAGTCGTTCAGTGTG
SnocOBP9M54A-R	CACACTGAACGACTCCGCCATGCAATGCATGTAAC
SnocOBP9F57A-F	CATTGCATGATGGAGTCGGCCAGTGTGATCGACGAGAAC
SnocOBP9F57A-R	GTTCTCGTCGATCACACTGGCCGACTCCATCATGCAATG
SnocOBP9F71A-F	GTGAAATCGAAGAAGAAACTGCTGTGGGATTCCTACCAGAAC
SnocOBP9F71A-R	GTTCTGGTAGGAATCCCACAGCAGTTTCTTCTTCGATTTCAC
SnocOBP9F74A-F	GAAGAAGAAACTTTTGTGGGAGCCCTACCAGAACAGTTTCAGAC
SnocOBP9F74A-R	GTCTGAAACTGTTCTGGTAGGGCTCCCACAAAAGTTTCTTCTTC
SnocOBP9L116A-F	CCACTTGCTCCAGAATTAGCGTACGTGTTG
SnocOBP9L116A-R	CAACACGTACGCTAATTCTGGAGCAAGTGG

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
