# Peer review of "Key Residues Affecting Binding Affinity of Sirex noctilio Fabricius Odorant-Binding Protein (SnocOBP9) to Aggregation Pheromone"

_ijms, 2022, doi:10.3390/ijms23158456_

Round 1
Reviewer 1 Report
This is very well organized paper. The only one thing I recommend are clear 2-3 conclusions except of lines 495-508.
Author Response
We would like to thank you for your careful reading, helpful comment, and constructive suggestion, which has significantly improved the presentation of our manuscript. The revisions are shown in file "Revise 1".

Reviewer 2 Report
The paper represents a significant amount of experimental and computational work that has the foundation to be a significant contribution to the field. However, as outlined below, there are a number of serious experimental concerns which must be clearly addressed in order to appropriately interpret the data. Overall, the paper could use additional proofreading as there are a number of incorrect or unusual phrasing/tense/word choices.
Specific concerns/suggestions:
1. The authors never define "OBP" as Odorant Binding Protein in the abstract/intro.
2. It is unclear why some text is blue.
3. FIgure 2 - the SDS page gel is inconclusive. It apears as if the band of interest is at/near the bottom of the gel, so it is impossible to tell if it also contains smaller sized fragments that are not resolved. A higher resolution gel designed to separate small MW proteins such as the protein of interest is needed, similar to what was shown for the western blot.
4. FIgure 3 - the MS is uninformative as presented. Without highlighting the expected ion fragments and how this was mapped to the protein sequence, the data is uninterpretable.
5. Figure 4 - the authors should use a more standard unit on the Y-axis such as molar ellipticity.
6. The NPN assay is not clearly explained. Where does NPN bind to the protein? What is the mechanism of competition? Is it direct competing for the same binding site? allosteric ejection of the probe upond ligand binding? Importantly, the alanine scan IMPLIES that the NPN binds to the pocket, but NPN fluorescence is so heavily environmentally sensitive, specifically sensitive to polarity of the environment, replacement of Met, Leu, or Phe dramatically reduces the local hydrophobicity and could impact NPN fluorescence.
There is definitely SOME effect of the volatiles on fluorescence, however this effect is not clearly binding dependent. The authors do not show any controls of the volatiles interacting with the fluorophore, i.e. a quenching effect. The quenching would be nonlinear similar to a binding curve (as observed) if the fluorophores were non-identical in their binding locations on the protein and/or if the probe is not fully exposed to the volatile molecules.
7. Figure 10 - The authors should show a different version in supplement. While the version of Fig 10 is very informative, the readers would also benefit from a "zoomed out" version to visualuize the location of the binding pocket with respect to the whole globular protein structure. This could be done using the same/similar models in Figure 7, however with the binding pocket clearly highlighted.
8. Figure 6 and 12 - fluorescence is spelled incorrectly in the figure axis.
Author Response
Thank you for your precious comments and advice. Those comments are all valuable and very helpful for revising and improving our paper, as well as the important guiding significance to our researches. We have carefully considered all comments from the reviewer and revised our manuscript accordingly. We hope that our responses could well addressed all concerns from the reviewer. The revisions are shown in file "Revise 2".

Round 2
Reviewer 2 Report
The authors did an outstanding job improving the manuscript in very short order.
Two suggestions remain:
1. I believe the authors should include the supplemental TABLE regarding the MS identification of the proteins in the main paper and the figure in the supplement. The table contains more useful info. Related, in the the authors should include a footnote regarding the "score" and how to interpret. Similarly, the authors should highlight the peaks in the MS spectrum with which peptides they correspond to.
2. The authors should add some sentence or sentences to highlight that the exact MECHANISM of the NPN binding/displacement assay is not elucidated. While the references show it's utility, the caveat of exact binding sites and mechanism of fluorescence decrease must be mentioned
Author Response
We thank the Reviewer for reading our paper carefully and giving the positive comments. Meanwhile, we would like to thank you for your careful reading, helpful comments, and constructive suggestions, which has significantly improved the presentation of our manuscript. The corresponding modifications were shown in the file "Revise 2_round2".

This manuscript is a resubmission of an earlier submission. The following is a list of the peer review reports and author responses from that submission.
Round 1
Reviewer 1 Report
Review ID ijms-1597222
Key residues affecting binding affinity of Sirex noctilio Fabricius odorant binding protein (SnocOBP9) to aggregation pher-3 omone
It is obvious that there is a growing interest in environmentally friendly methods of plant protection worldwide. Large number of pesticides that are ineffective and have negative impact on the environment have been reduced. In present time, environmentally friendly methods of plant protection are of importance where natural defense mechanism of plants based on volatile organic compounds may play significant role.
This is quite well organized manuscript. I found this “ms” interesting and innovative. However, a few questions must be explained more precisely.
Critical review:
- I am surprised to see that (Z)-3-decen-ol was identified by you as male pheromene. In the absence of a choice, any component can be a choice. Please answer this sentence.
- I don’t see any clear conclussions.
- Lines 492-494. I don’t undesrstand this section.
- How do you know that males were unmated? There is large difference in respond between unmated and mated insects of both sexes. What about mated individuals?
- Lines 487-494. The whole section is rather poorly described.
- The rest of the manuscript is well described. Figures are well presented too.
One other paper to add:
Sitophilus granarius responses to blends of five groups of cereal kernels and one group of plant volatiles. Journal of Stored Products Research 62: 36-39.
DOI: 10.1016/J.JSPR.2015.03.007
Reviewer 2 Report
This remains one of these very tedious studies which consist inchoosing one gene in the insect odorant-binding protein (OBP) family,
preferably from a local pest, eventually related to agricultural or
forest industry in China, express it in a recombinant system (not
adapted to soluble extraction) and perform some binding studies using
fluorescence spectrometry assay. Knowing that this technology can
generate many fake or false positive (and negative) results, the
binding results cannot be accepted when using only a few chemicals
(similar structures) and not confirmed by other more rigorous assay.
The text per se remains also not rigorous enough for publication,
Hymenopetera instead of hymenoptera (line 14), insects pheromone
volatile, and so on. Twelve figures on the raw + 5 tables (some have
only one line) and tons of mathematical formula are rather
unacceptable for a decent readable relevant publication. More efforts
should have been made to show a very clear readable result, if any.
Most of the figures presented here (SDS PAGE of recombinant proteins,
no blotting or further analysis of protein identity, and binding curve
for the probe with scatchard plot, structural modeling from low
template identity) are supplementary materials and certainly they
would be considered as such in many and most of all international
journals with very decent and relevant impact.
Most of the figures presented here remain extremely poor, see figure 1
which is completely unreadable, what sequence has been expressed (from
first methionine to C-terminal residue), how was the clone been found,
which tissue and developmental data are in support with a function
about pheromones? Tissue-developmental data should be provided before
to insist about the olfactory function of OBP9.
Again, the authors should read the complete and recent literature on
insect OBPs. There is growing evidence that these proteins play no
role in olfaction, this should be strongly considered before to
introduce OBP and investigate functional properties, especially when
using such a short set of ligands, only limited and restricted to
putative odor chemicals, pheromones and quasi-identical structures.
They cite Guo et al. Frontiers in Physiology (2021): “Although the
functions of OBPs are currently considered to be diverse (not diverse,
not tuned to olfaction), especially when they have no-antenna-specific
expression (what does this mean?). Such as, a new study in Bombyx mori
showed that the investigated OBPs (pheromone binding proteins and
general odorant binding proteins that gave the name to the OBP family)
are expressed in non-olfactory tissues such as the gut and fat body at
many different developmental stages, strongly indicating that these
OBPs (and all other OBPs) have non-olfactory functions (see lines
50-53). It should be mentioned that Guo et al. showed preferential
binding to vitamins compared to odor chemicals, and this seriously
affects the goal of present study, which completely neglects to test
chemicals outside the olfactory paradigm. This is required especially
because the authors completely neglect to show tissue distribution
data, which is a strong limitation for publication after Guo et al.
(2021) and forty years of debate on the OBP function.
OBPs are found in the gut and fat body, so no role in olfaction. Here,
it is now obvious that OBPs play NO olfactory function. Here, the
authors should make efforts, especially for IJMS, to describe the
tissue and developmental pattern of SnocOBP9 before to babble about
the function. The function to olfaction is probably unreliable if the
gene is NOT specifically expressed in the antennae. Please provide
clear distribution of gene expression, when and where, before to tell
again about antennal-specificity and intend to report about odor
function. Structure and binding data should agree with physiological, tissue distribution and developmental data.
In the oriental leafworm moth, Zhu et al. reported 87% of chimeras in
surviving injected embryos and adults. Homozygous SlitPBP3 knockout
mutants were tested for their response to three main sex pheromone
components using electro-antennography technique (EAG). The antennal
responses persisted in PBP3-mutants, suggesting a minor role for this
protein in pheromone perception, if any. Zhu GH, Xu J et al. (2016)
Functional characterization of SlitPBP3 in Spodoptera litura by
CRISPR/Cas9 mediated genome editing. Insect Biochem Mol Biol 75: 1-9
is still excluded from this nieme version.
REJECT
The present set of data and long list of figures (which remain to be
moved to supplementary materials when confirmed) should be rejected
for additional and too many other critical points that remain:
1) It is not clear about the biological relevance of the ligands
tested. Unsaturated short-chain alcohol, (Z)-3-decen-1-ol, highly
volatile compound, no contact pheromone, pheromone (for aggregation)
in wasps? The authors cite “Biology of a putative male aggregation-sex
pheromone in Sirex noctilio (Hymenoptera: Siricidae) Guignard et al.
2020 PLoS ONE, and mention that these two new substances (?) were not
identified in their previous experiments”. This discrepancy and
failure in identifying the complete mixture of pheromones is matter of
debate. It should be discussed, because it rather conflicts the
objectives of the authors that target one OBP and a ligand with
obscure function. “These results (which ones?) indicate that Z3D may
plays (?) an important role in regulating the behavioral rhythm of
woodwasps” is rather inappropriate statement (see line 115).
2) As previously mentioned, introduction should refer to non-olfactory
function when refers to OBP, particularly moth PBP and GOBP. SlitPBP3
has been demonstrated to have no role in olfaction, and this is still
excluded from Introduction, Results and Discussion.
3) The introduction and prospect about OBPs are turned to olfaction,
odorant receptors (OR) and other gene families usually described in a
purpose of studying olfaction. This is not the case here. While OR is
specifically expressed in antennal sensilla, there is by far a strong
lack of knowledge to correlate OR expression with the investigated
OBP. The OBP9 expression in the wasp antennae needs to be proved (see
line 39: no proof for this statement). OBPs including PBPs and GOBPs
mainly express in metabolic organs when insects are exposed to
chemical stress (Guo et al., 2021).
4) Tissue (and development) data are missing not only here, but also
in Guo B. et al. 2019, 2021, which only reported about antennal
transcriptome, no clear tissue distribution analysis, which should be
shown and properly analyzed to conduct further analysis with odors,
including docking.
5) The protein seems to be obtained in a good purity quality. Much
better and pure than some other works published at IJMS, indeed. Eventually the figure of gels should be enlarged.
However, the critical point here remains that proteins have been
expressed/obtained in inclusion bodies of bacterial cells. This means
they lost appropriate configuration, in inclusion bodies recombinant
proteins fail to be in a proper functional folding. From unfolded,
they should be refolded before to carry out binding studies. Please
provide proofs, controls and refolding data to clearly insure and
demonstrate that the protein samples in use are tested in a proper
functional conformation.
6) In addition, please provide data about protein identity. Only the
apparent molecular weight on SDS-PAGE is not enough to guarantee about
the identity of the protein expressed. This requires OBP-immunoassay,
mass spectrometry, LC-MS-MS, and/or N-terminal sequencing. These data
should be provided here to attest of the protein identity. One protein
or variant (SnocOBP9L116A) does not seem to have the proper size for
OBP (less than 12 kDa). How can this be explained by one-to-one amino
acid mutation?
7) Do these OBP9s express naturally in bacteria? How do the authors
explain such a low protein concentration in the bacterial cells
induced by IPTG (line 2) for SnocOBP9F71A and SnocOBP9F74A compared to other samples (Figure 4)?
8) Please provide the yield obtained for each recombinant protein in
expression system? It does not seem that the variants are equally
expressed, and some variants do not express in amounts compatible with
several replicates. Please provide data and statistical analysis for
n=9.
9) SDS-PAGE (figure 2), binding curve of 1-NPN for reference (figure
3) and “chemical structure” (figure 4, no legend) are rather
inappropriate figures in the main text. The curves on figure 5 do not
show selective binding for Z3-decen-1-ol. See table 1: binding value
also high for limonene. Usually a set of 20-50 chemicals is required
to provide conclusive results. More ligands needs to be tested to
valid the results prone for Z3-decen-1-ol. The volatility of the
ligand should also be considered, limonene, Z3 and decen-1-ol are
extremely highly volatile and I strongly doubt there can be anything
like a reliable data if the experiment has not been done at 4C. How
was the ligand solution prepared? What is the starting concentration
of the ligand? What is the concentration of the ligand even after a
few seconds-exposure in quartz tube?
10) Six different OBP sequences corresponding to mutants have been
expressed. Pheromone binding data should be provided for the six OBP
mutants to attest of “key amino acid residues affecting binding
affinity of Sirex noctilio odorant binding protein (SnocOBP9) to
aggregation pheromone”.
11) ASPs? Is it a new class of OBPs? “AmelASP1 had the highest
similarity (44.92%) to the sequence identity of SnocOBP9, which
indicated that AmelASP1 could be used as a homology modeling template
for SnocOBP9.” NO. NOT AT ALL. Nothing reliable can be said in
homology modeling using a reference with <50% homology to test
protein. In addition, the choice of template is not judicious (say
wrong) if we consider the biological relevance of the cognate ligands.
ASP1 has been described as queen pheromone (9-ODA/9-HAD)-binding
protein in the bee Apis mellifera. Is this protein expressed in the
mandibular/Dufour’s gland that produces the queen pheromone? Is
SnocOBP9 produced in the mandibular/Dufour’s gland of the wasp? In
which glands or tissues are SnocOBP9 and variants expressed? If the
tissue profiling is different, probably the function is different and
certainly the two proteins cannot be superimposed. This is matter of
rejection not only for IJMS, but also for any international journal of
decent envergure. The two protein sequences should be aligned before
to set out structural modeling, this may explain such high discrepancy
observed in the N-terminus (see Figure 6). A protein involved in
binding bee pheromone such as 9-oxo-2-decenoic acid (9-ODA), both
enantiomers of 9-hydroxy-2-(E)-decenoic acid (9-HDA), methyl
p-hydroxybenzoate (HOB), and 4-hydroxy-3-methoxyphenylethanol (HVA)
certainly retains a different functional pocket than an OBP turned to
aggregation pheromones such as Cis-3-decenol. The story remains not
rigorous here, before to extrapolate a series of wrong and hasty
conclusions. It should be told somewhere here whether ASP1 and
SnocOBP9 display the same binding pocket, or some or a few similar key
amino acids in their respective binding pocket.
12) Where is the binding pocket on the protein models shown on Figure
6? In any case, a more robust analysis of protein structure (Linux,
NMR, X-Ray) is necessary to perform a reliable study about specific
functional elements. Time-evolution RMSD curves (Figure 7), Cluster
analysis (table 2), binding free energy scores for one single line of
protein or complex (table 3), unlegended and raw RMSF curves of a
protein sequence without indication on the binding pocket and N-tail
(figure 10), the decomposition of residues for which interaction or
type of interaction with ligand is dubious (table 4), how can Met54 so
distantly connected to the ligand can contribute the same total energy
than Phe57 in direct contact with the ligand? The values are not so
different than that measured for Leu116 which fell off from the
binding site (see table 4 and Figure 9). This set remains incoherent.
13) Same remark for figure 11 and table 5: equilibrium conformation,
scanning, similar results for Leu116, Met54 and Phe57? This conflicts
the position of the amino acid residues displayed on figure 9 on this
version and on the early version of the manuscript.
14) The position (distance in angstrom) for each ligand-residue
should be indicated.
15) There are different scores and ligand poses to test before to
validate a docking result.
16) The surface view of the hydrophobic cavity of Z3D anchored in
SnocOBP9 could be the only figure of interest (Figure 8), but this
figure is poorly legended. The alpha-helix profile is completely
different than the profile shown on other computered structural
modeling figures (6B and 9). Then, it is rather unclear what the
authors design when they refer to “hydrophobic pocket”. Z4D is
considered to be a fatty alcohol, very hydrophobic molecule,
practically insoluble in water, so is it the same situation with Z3D
that can shape in fused ring? Conserved Phenylalanine(s) are known to
be important to stabilize double bond. Is it the case for the Phe(s)
residues of SnocOBP9 protein? How is the hydroxyl group of Z3D
stabilized in the SnocOBP9 binding pocket? Ligand-OH does not bind to
anything on the protein model shown on Figure 9. It can be understood
since the binding pocket tested (ASP1) is turned to accommodate long
straight hydrophopic chains of decenoic acid, but not so short potentially
cyclic molecules such as Z3-decenol.
17) The state after mutation (table 4) is destabilizing for the five
amino acid residues tested, but mutation energy does not differ much
between Met54, Phe57 and Leu116, which conflicts with their respective
position in the protein model shown on figure 9. Phe57, but not Met54
and Leu116 seem to contact the ligand. What does SnocOBP9-Z3D mean?
CpomPBP2-Dod (which even did not correspond to the purpose here) has
been removed from Table 2 on an early version. It is really confusing
now as whether the values reported on Table 2 deals with the
investigated protein or another protein on which similar or same
mistakes and wrong assumptions have been carried out. Are the authors
very sure about the origin of data for S. noctilio and OBP9 analysis?
The results presented before seemed to be a mix between different
species and OBP-ligand complexes.
REJECT
18) On Figure 12, data in B should follow data in A. Please show 1/Ki
measurement for all mutants.
19) When the binding properties of a shorter peptide such as
SnocOBP9L116A (see Figure 2) is significantly altered by mutation (see
Figure 12A) when the residue is not crucial for Z3C ligand interaction
(see Figure 9), the results rather become extremely odd, particularly
when mutation does not alter the binding properties of Phe57-mutant
(see Figure 12A).
20) One control in further modeling experiment should be the
positioning of the fluorescent probe 1-NPN (1-N –phenylnaphthylamine).
This would allow tracing what the authors measure in quartz and what
the authors measure in silico.
21) No other conclusion than a simulation can be drawn from the
molecular dynamics simulations or computational scanning described
here in the material and methods and used to build the story.
REJECT
Round 2
Reviewer 2 Report
The work presented here is not complete and rigorous enough to take
obvious part in the OBP debate.
Because incomplete and not rigorous enough, the data fail to be conclusive on the functional relevance of SnocOBP9, particularly in relation with odor and pheromone aggregation.
My major concern remains: the authors did not provide a relevant
tissue-distribution analysis (analysis of abdomen sample does not mean
analysis of dissected pure Dufour’s gland sample), which is strongly
required to debate about the function. Binding to pheromone
aggregation needs evidence of expression in Dufour’s gland and/or hind
legs. Do the authors report OBP9 expression in hind legs? Tendon
gland? (see reply to comments page 14).
They repeated or re-performed qPCR experiments (Does it mean that the
authors use another reference gene?). They are still limited to a few
tissues (see Figure 2). The amounts of relative expression are
gigantic (average value of 60000-40000 means standard deviation value
very high), which is rather dubious. Then, there are no statistical
differences between male and female, so it is rather difficult to
argue for SnocOBP9 tuned to male pheromones.
Another element ofrejection is the lack of data for Day-1. See Figure 3: FA1 is missing. The data for one-day-old adults is required for rigorous analysis
comparing Fig. 3 and Fig. 5B. There is little mating activity at day-1, similarly to day-5 and day-6, so we expect to see similar levels of OBP9 comparing FA1 and FA5-FA6. Data (FA1 and MA1) are missing here (see Fig. 3).
On Figure 3, relative expression values exceed 100000, SD error about 10000-20000, how is that possible? The value results are different than those on Figure 2. The OBP9 gene expression profiling (Fig. 3) contrasts with mating activities (Fig. 5): linear decrease of mating activity after day-2, no statistical differences in gene expression between day-2 and day-3, FA4-FA7 and MA4-MA7. Therefore, there is no correlation between OBP9 gene
expression and mating.
Figure 4 should be removed. It only shows strong sexual dimorphism in
size and color in Sirex noctilio, which may explain the differences
observed between males and females.
Another key point of criticism is that the authors did not test
binding to non-semiochemicals, this is required after Guo et al.
(2021). The mystery of OBPs has remained for 40 years, due to no
efforts to link binding data and insect physiology, and taking for
granted that OBPs play a role in olfaction, due to their high
abundance in the insect antennae (and legs). This is true for most of
the articles cited here, too many IJMS, true, including the OBPs of
the spider mite, which fails to support the functional argument not
only by immunoassay, mass spectrometry, LC-MS-MS and/or N-terminal
sequencing, but also by a complete omission of where and when the gene
is expressed. Similarly, pest management science is not compatible with protein biochemistry work, which needs to evaluate the proper folding of the protein and test a significant amount of ligands.
The modeling of OBP9 based on bee ASP1 (different sequence and
different function) is still dubious. In any case, it should analyze
binding to non-semiochemicals as controls before to make a conclusion
about odors. Many patch-clamp studies recording neuronal activities in
absence of OBPs have shown that OBPs are not required for the cell
response and odor detection in invertebrates and vertebrates. More,
the time delay of neuronal response is lower that the binding kinetics
measured for OBP-ligand. All together, it leaves a serious doubt on
the olfactory/pheromone binding function of insect OBPs.